# Prevalence and Factors Associated with Cardiovascular Lifestyle Risk Factors among Pregnant Women in Burkina Faso: Evidence from a Cross-Sectional Study

**DOI:** 10.3390/ijerph20010102

**Published:** 2022-12-21

**Authors:** Franck Garanet, Abou Coulibaly, Adama Baguiya, Fati Kirakoya-Samadoulougou, Seni Kouanda

**Affiliations:** 1Département Biomédical et Santé Publique, Institut de Recherche en Science de la Santé (IRSS), Centre National de la Recherche Scientifique et Technologique(CNRST), Ouagadougou 03 BP 7047, Burkina Faso; 2Laboratoire de Santé Publique, Ecole Doctorale Science de la Santé (ED2S), Université Joseph Ki-Zerbo, Ouagadougou 03 BP 7021, Burkina Faso; 3Centre de Recherche en Epidémiologie, Biostatistique et Recherche Clinique Bruxelles, Ecole de Santé Publique, Université Libre de Bruxelles, 1070 Brussels, Belgium; 4Institut Africain de Santé Publique, Ouagadougou 12 BP 199, Burkina Faso

**Keywords:** prevalence, cardiovascular lifestyle risk factors, pregnancy, Burkina Faso

## Abstract

Data on lifestyle risk factors for cardiovascular diseases, such as smoking, alcohol consumption, inadequate physical activity, and insufficient consumption of fruits and vegetables, in pregnant women in Africa, are scarce. This study aimed to estimate the prevalence of cardiovascular lifestyle risk factors among pregnant women in Burkina Faso and identify their associated factors. Pregnant women who attended antenatal care services between December 2018 and March 2019 were included in this study. A modified Poisson regression model was used to estimate adjusted prevalence ratios (aPR) with a 95% confidence interval. A total of 1027 pregnant women participated in this study. The prevalence of alcohol consumption, tobacco use, inadequate physical activity, and insufficient consumption of fruits and vegetables were 10.2% [8.5–12.2], 3.0% [2.1–4.3], 79.4% [76.9–81.8], and 53.5% [50.5–56.6], respectively. The prevalence of more than one cardiovascular lifestyle risk factor in pregnant women was 50.9% [48.0–54.0]. The predictors significantly associated with more than one cardiovascular lifestyle risk factor were women over 30 years old aPR 1.26; 95% CI [1.03–1.53]), women living in fourth wealth index households (aPR 1.23; 95% CI [1.06–1.42]), in semi-urban areas (aPR 5.46; 95% CI [4.34–6.88]), and women with no occupation (aPR 1.31; 95% CI [1.18–1.44]). The prevalence of more than one cardiovascular lifestyle risk factor was high during pregnancy in Burkina Faso. Women of childbearing age should be advised on how healthy behaviors can lead to improved pregnancy outcomes.

## 1. Introduction

Cardiovascular diseases (CVD) are among the leading causes of death worldwide [1]. In 2012, an estimated 17.5 million people died from CVD, 30% of all deaths worldwide; 7.5 million were due to coronary heart disease, and 6.7 million died from a stroke. By 2030, it is estimated that 23.6 million people will die annually from cardiovascular disease and stroke. These conditions are projected to remain the leading cause of death [2]. 

Smoking, alcohol consumption, inadequate physical activity, and the insufficient consumption of fruit and vegetables are well-known cardiovascular lifestyle risk factors [3,4,5,6]. Worldwide, smoking is responsible for 10% of deaths attributed to cardiovascular diseases, gradually increasing in resource-limited countries. Alcohol consumption contributes to 6% of deaths among men and 1% among women. Inadequate physical activity and the insufficient consumption of fruits and vegetables contribute to 33% and 10% of ischemic heart disease-related deaths, respectively [3,7,8,9]. For example, in Burkina Faso, the World Health Organization Stepwise approach to non-communicable disease (NCD) risk factor surveillance survey 2013 (STEPS) revealed a national prevalence of smoking of 30.8%, alcohol consumption of 17.4%, inadequate fruit and vegetable consumption of 98.9%, and insufficient physical activity of 3.5% [10].

Cardiovascular diseases during pregnancy include congenital heart disease, valvular heart disease, ischemic heart disease, peripartum cardiomyopathy, hypertensive disorders, and venous thromboembolism. In addition, the global burden of cardiovascular disease during pregnancy has increased alarmingly in parallel with the increasing prevalence of lifestyle risk factors among women of childbearing age [11,12,13,14].

Cardiovascular diseases, considered diseases that affect the rich, are increasingly affecting the underprivileged population [15,16,17,18,19]. It is essential to address the risk factors for cardiovascular disease in pregnant women. Such efforts will help reduce adverse perinatal outcomes, such as low birth weight, fetal and infant mortality, and potential birth defects and cardiovascular disease in mothers [12,15,16,19]. Assessing the lifestyle risk factors for cardiovascular diseases could contribute to global prevention efforts. Five studies examined the risk factors for cardiovascular disease during pregnancy, such as smoking, alcohol consumption, low fruit, and vegetable consumption, daily salt intake, overweight or obesity, and hypertension [11,15,16,17,18]. However, few studies have identified the predictors of cardiovascular lifestyle risk factors in pregnancy [20,21]. Therefore, this study aimed to determine the prevalence of, and factors associated with cardiovascular lifestyle risk factors in pregnant women in Burkina Faso.

## 2. Materials and Methods

### 2.1. Study Setting

The study was conducted in Sanmatenga Province, North Central Burkina Faso. There are three health districts (Kaya, Boussouma, and Barsalogho). Kaya (the regional capital) is 100 km from Ouagadougou. We selected the three largest health centers in the districts of Kaya and Boussouma that received the highest number of women who attended antenatal care (ANC). These health centers are the health centers of sectors 4 and 6 and the medical center of sector 1. In addition, in rural areas, we selected the three largest health centers that received the most significant number of pregnant women who attended ANCs (Boussouma Medical Center, the Pissila health center, and the Korsimoro Medical Center).

### 2.2. Study Design 

We conducted a cross-sectional study from 15 December 2018 to 15 March 2019. The study population consisted of pregnant women, irrespective of their gestational age, who sought antenatal care at these health centers. All women who attended antenatal consultations were included in each health center during the collection period.

### 2.3. Sample Size

With a 50% proportion of pregnant women with cardiovascular lifestyle risk factors (unknown prevalence), a power of 80%, a precision of 0.05, a prevalence ratio of 1.5, and a nonresponse rate of 10%, the minimum sample size required was 900 participants. Therefore, we used the formula proposed by Kelsey et al. [22].
$$n1=\frac{\left(Z\alpha /2+Z1-\beta \right)2\;PQ\left(PR+1\right)}{PR\left(p1-p2\right)\dot{2}}$$
And *n*2 = *PR***n*1;

where; 

*n*1 = number of exposed; *n*2 = number of unexposed;

Zα/2 = standard normal deviate for a two-tailed test based on alpha level (related to the confidence interval level); Z1 − β = standard normal deviate for a one-tailed test based on beta level (related to the power level);

PR = ratio of unexposed to exposed; 

*p*1 = proportion of individuals exposed to disease and *q*1 = 1 − *p*1; *p*2 = proportion of unexposed individuals with disease and *q*2 = 1 − *q*2;
$$Q=\frac{p1+PRp2}{PR+1}$$
And *Q* = 1 − *P*.

### 2.4. The Variables in the Study

The main outcome variables were cardiovascular lifestyle risk factors, including fruit and vegetable consumption, physical activity, tobacco use, and alcohol consumption. First, we considered these outcomes separately and then computed another binary outcome variable that identifies the presence of at least two risk factors. 

Cardiovascular lifestyle risk factors were operationalized based on standard definitions to ensure comparability and minimize errors. Participants who had consumed any form of alcohol or tobacco (with or without tobacco smoking) in the past 30 days were considered current users. Women who used tobacco daily were defined as current tobacco users. We used “exposure cards” associated with the food frequency questionnaire to collect data on the consumption of fruits and vegetables during pregnancy [23]. Fruit or vegetable intake was considered inadequate if the participant consumed fruit or vegetables less than five times daily. According to the World Health Organization (WHO), consuming five servings of fruits and vegetables daily is associated with a lower risk of death from cardiovascular disease, cancer, and respiratory disease [23].

We adopted the WHO Global Recommendations on Physical Activity for Health (adults) to measure physical activity levels in pregnant women. Physical activity includes travel, recreation, and work. WHO-recommended moderate-intensity physical activity (at least 150 min per week) can help reduce the risks mentioned above in adults [24]. In our study area, women generally engage in household chores and agricultural activities that meet the criteria for moderate-intensity physical activity [25]. Participants were asked various questions about their level of physical activity. Based on their responses, the interviewer recorded the hours spent on daily household activities, travel, recreation, and work. Based on the hours spent engaging in various activities, the researchers then grouped these activities into two categories: (i) less than 150 min per week of moderate-intensity physical activity (physical inactivity) and (ii) more than 150 min per week of moderate-intensity physical activity (active physical activity). 

The independent variables were age, level of education, socioeconomic status (SES), geographical location, and occupation. Another variable was pre-existing illnesses or a family history of cardiovascular diseases (hyperglycemia and diabetes, high blood pressure, and a family history of hypertension).

### 2.5. Data Collection

Data were collected on the day of each participant’s consultation. A room was identified at each site, and each consenting woman was interviewed. A structured questionnaire was used for data collection. Health workers with experience in data collection and who spoke the local language were recruited to collect data. After face-to-face interviews to obtain demographic characteristics, maternal smoking, alcohol consumption, and blood pressure were measured. An electronic blood pressure monitor (Omron M3 Intellisense device) with an adult-sized cuff was used. Blood pressure was measured with the participant seated at the end of the interview after approximately 15 min of rest. Blood pressure was measured in each arm to determine the arm with the highest blood pressure. Two other measurements were successively performed on the identified arm at an interval of one minute. The average of the last two measurements was considered as the participant’s blood pressure.

### 2.6. Data Quality Assurance

The research team trained all health workers in the use of tools and different data collection techniques. A pre-test was conducted at a health center that was not included in the study. A manual covering all the required interview procedures was written and provided to each field worker. 

### 2.7. Data Analysis 

All statistical analyses were performed using Stata V.14.1 (Stata 2014: Revision 1 December 2015. College Station, TX, USA) [26]. A descriptive analysis was performed, and the variables were recorded as required. 

For SES, 20 items related to household assets were used with principal component analysis (PCA) to generate wealth quintiles (Poorest, Second, Middle, Fourth, and Richest) [27]. These 20 items included the presence of a bucket, bowls, cup, gas fireplace, bed, mattress, table, chair, functional radio, functional clock, functional lamp, functional television, functional bicycle, functional motorcycle, functional telephone, functional cart, functional wheelbarrow, donkey or horse, and poultry, sheep, or goats. We then produced a wealth quintile.

The variables used for this study were recorded as follows: age (≤19 years, 20–29 years, and ≥30 years), educational level (one, primary, and more), wealth quintile (lowest, second, middle, fourth, and highest), geographical location (semi-urban and rural), occupation (employed or not employed), parity (primipara or multipara), marital status (married or not married), pre-existing illnesses, or history of hypertension (yes or no).

A modified Poisson regression model, in order to generate a prevalence ratio with a 95% confidence interval, was fitted to identify factors associated with cardiovascular lifestyle risk factors [28,29].

## 3. Results

### 3.1. Characteristics of the Participants

A total of 1027 pregnant women participated in this study. The mean age was 25.8 ± 6.0 years. One hundred and forty-eight (14.4%)participants were under 20 years of age. Most study participants were aged 20 to 29 (61.6%). For most women, 63.3% did not attend school. 42.6% of the women live in rural areas. Overall, 4.6% of women had pre-existing illnesses or a family history of hypertension (Table 1). 

### 3.2. Prevalence and Associated Factors of Alcohol Consumption

The prevalence of alcohol consumption was 10.2% (95% confidence interval (CI) [8.51–12.23]). This prevalence was high in the not employed group (10.9%) and the fourth wealth quintile group (13.2%) (Table 2).

In the multivariable regression model, the predictors significantly associated with alcohol consumption were primary or higher education level (adjusted Prevalence Ratio (aPR) 1.82; 95%CI [1.27–2.62]) and rural area (aPR) 1.94; 95% CI [1.31–2.87] (Table 2).

### 3.3. Prevalence and Associated Factors of Tobacco Use

The prevalence of tobacco use was 3% [2.1–4.3]. This prevalence was higher in the over 30 years group (7.7%), among those living in rural areas (6.6%), and among women with pre-existing illnesses or a family history of hypertension (6.4%) (Table 3).

In the multivariable regression model, the predictors significantly associated with tobacco use were: the 20–29 years group (aPR 0.15; 95% CI [0.04–0.50]), women in rural areas (aPR 15.02; 95% CI [3.12–72.39]), and multipara (aPR 10.07; 95% CI [3.38–29.14]) (Table 3).

### 3.4. Prevalence and Associated Factors of Inadequate Physical Activity

The prevalence of inadequate physical activity was 79.4% [76.9–81.8]. This prevalence was higher in semi-urban areas (93.9%) and employed women (86.1%) (Table 4).

In the multivariable regression model, the predictors significantly associated with inadequate physical activity were the second, middle, fourth, and highest wealth indices. Pregnant women with high wealth indices (middle, fourth, or highest) were more likely to have inadequate physical activity than those with lower wealth indices. Women living in semi-urban areas were 1.54 times more likely to have inadequate physical activity compared to women living in rural areas (aPR 1.54; 95% CI [1.42–1.66]) (Table 4).

In the multivariable regression model, the predictors significantly associated with inadequate physical activity were the second, middle, fourth, and highest wealth indices. Women living in semi-urban areas were 1.54 times more likely to have inadequate physical activity compared to women living in rural areas (aPR 1.54; 95% CI [1.42–1.66] (Table 4).

### 3.5. Prevalence and Associated Factors of Insufficient Consumption of Fruit and Vegetables

The prevalence of insufficient consumption of fruit and vegetables was 53.5% [50.5–56.6]). This prevalence was 62.2% in the ≤19 years group and 94.7% in rural areas (Table 5).

In the multivariable regression model, the predictors significantly associated with alcohol consumption were women with no education level (aPR 1.16; 95% CI [1.03–1.30]), the highest wealth group (aPR 1.19; 95% CI [1.03–1.40]), the rural area (aPR 4.47; 95% CI [3.85–5.18]), the employed group (aPR 0.73; 95% CI [0.66–0.80]), and the Pre-existing illnesses or family history of hypertension (aPR 1.17; 95% CI [1.03–1.33]) (Table 5).

### 3.6. Prevalence of More Than One Cardiovascular Lifestyle Risk Factor and Its Associated Factors

The prevalence of more than one cardiovascular lifestyle risk factor in pregnant women was 50.9% (95% CI [47.9–54]). The predictors significantly associated with more than one cardiovascular lifestyle risk factor in the multivariable regression model were: women over 30 years old (aPR 1.26; 95% CI [1.03–1.53]), women living in fourth wealth index households ((aPR) 1.23; 95% CI [1.06–1.42]), in semi-urban areas (aPR 5.46; 95% CI [4.34–6.88]), and women with no occupation (aPR 1.31; 95% CI [1.18–1.44]) (Table 6).

## 4. Discussion

This study estimated the prevalence and predictors of four main cardiovascular lifestyle risk factors in pregnant women: alcohol use, tobacco consumption, inadequate physical activity, and insufficient consumption of fruits and vegetables. This study is the first to model cardiovascular lifestyle risk factors in pregnant women in rural and semi-urban areas with low resources.

### 4.1. High Prevalence of Inadequate Physical Activity and Insufficient Consumption of Fruits and Vegetables

The most common cardiovascular lifestyle risk factor in pregnant women is physical inactivity, with a prevalence of 79.5%, which is higher than the prevalence reported in a survey of risk factors for non-communicable diseases in the general population in Burkina Faso (70.7%) [10]. Paudel et al., in a community-based cross-sectional study of 426 participants conducted in Nepal in 2018, showed a prevalence of 53.9% [20]. This may be explained by the general belief among the population (promoted by some local healthcare providers) that physical activity leads to undesirable pregnancy outcomes such as miscarriage, poor fetal growth, or premature delivery [30,31,32]. Additionally, semi-urban and rural areas have no space for physical activity. However, a systematic review of physical activity during pregnancy found that some mild to moderate physical activities protect maternally and child health from outcomes such as preeclampsia, gestational hypertension, and preterm birth [24]. Global physical activity guidelines recommend moderate-intensity exercises such as brisk walking and other recreational activities for pregnant women [33]. Therefore, providing appropriate advice to encourage optimal physical activity during pregnancy may reduce cardiovascular lifestyle risk factors.

After physical inactivity, the most common risk factor was the insufficient consumption of fruits and vegetables in pregnant women, with a prevalence of 53.6%. The high proportion of pregnant women who do not consume enough fruits and vegetables could be due to poverty, which could prevent them from being able to afford fruits and vegetables. Other studies have argued that low income prevents individuals from purchasing sufficient amounts of fruits and vegetables [25,34]. Another potential factor associated with lower consumption of fruits and vegetables during pregnancy may be their seasonal and geographic availability. In most households in Burkina Faso, fruit consumption is not considered a priority compared to the daily consumption of the main cereals, especially rice and sorghum [35].

### 4.2. The Prevalence of Alcohol and Tobacco Consumption

The prevalence of alcohol (3%) and tobacco consumption (10.2%) among pregnant women are other key findings of this study; Paudel and al., in Nepal, have shown a prevalence of alcohol consumption (13.3%) and tobacco use (21.3%) [20]. In pregnant women, tobacco is often consumed by chewing tobacco to combat nausea, especially during the first trimester of pregnancy. However, healthcare providers have overlooked this issue. The reason for alcohol consumption among pregnant women could be the widespread availability and accessibility of traditional alcohol and imported alcoholic beverages, even in rural areas.

Another common reason for alcohol consumption in pregnant women could be that several types of traditional alcoholic beverages are brewed at home in our study area, most often by women. In most ethnic groups, “dolo,” a local beer, is considered culturally acceptable even for pregnant women. This may be due to a lack of knowledge regarding the harmful effects of these products on the mother and fetus during pregnancy [35]. Therefore, health promotion strategies should include general awareness programs to reduce cardiovascular lifestyle risk factors.

### 4.3. The Co-Occurrence of Cardiovascular Lifestyle Risk Factors and Associated Factors

Our study reported that half (50.9%) of pregnant women had more than one cardiovascular lifestyle risk factor.

Women who lived in semi-urban areas were five times more likely to have more than one cardiovascular lifestyle risk factor than those who lived in semi-urban settings. Unemployed women were 1.31 times more likely to have more than one cardiovascular lifestyle risk factor than employed women. Thus, after adjusting for other potentially confounding variables, our study demonstrated an association between rural settings, unemployment, and cardiovascular lifestyle risk factors. Adjusting for potential confounding factors yielded more direct evidence of the contribution of these parameters to the high prevalence of cardiovascular lifestyle risk factors in pregnant women. The prevalence of these factors may be due to a lack of awareness about risky behaviors during pregnancy and cultural factors.

Similarly, limited access to suitable fruits and vegetables may be associated with poverty in the study area [25,34]. According to the World Bank, more than 40.1% of the population lives below the national poverty line [20]. Pregnant women living in the fourth wealth index were 1.23 times more likely to have more than one cardiovascular lifestyle risk factor than those living with a low wealth index.

Several studies have provided results consistent with our findings [15,36,37]. For example, in 2018, Paudel et al. demonstrated that a high wealth index was significantly associated with cardiovascular lifestyle risk factors [20]. In 2007, Dumith et al. demonstrated that educational level was significantly associated with physical inactivity [30]. Therefore, targeted education campaigns and poverty reduction strategies should be recommended to reduce the risk factors for cardiovascular lifestyles in pregnant women.

Additionally, any contact with mothers should be an opportunity to encourage them to engage in activities that prevent cardiovascular lifestyle risk factors in pregnant women.

### 4.4. Limitations

This was a hospital-based, cross-sectional study. Although we ensured confidentiality during the interviews, this study was subject to recall and social desirability biases (particularly for tobacco consumption), which may underestimate or overestimate the prevalence.

We also did not consider the vegetables consumed in meals made of grains and soups. Furthermore, we explored the factors associated with lifestyle risk factors in this cross-sectional study. However, these associations can be better established by a more appropriate study design and robust analysis to provide evidence in the context of our study.

## 5. Conclusions

This study showed a high prevalence of low fruit and vegetable consumption and inadequate physical activity. Further emphasis on improving the management of these risk factors before and during pregnancy is critical for improving maternal and child outcomes and reducing disparities in adverse pregnancy outcomes. Any contact with pregnant women should be used as an opportunity to advise women on reducing cardiovascular lifestyle risk factors to improve the chances of a better pregnancy experience and desired outcomes.

However, existing recommendations for the prevention of cardiovascular lifestyle risk factors remain limited, and significant gaps in longitudinal, intrapartum care exist, leading to missed opportunities for the prevention of cardiovascular lifestyle risk factors in pregnancy. Further research is needed to determine optimal prevention strategies during and after pregnancy to improve women’s cardiovascular health.

## Figures and Tables

**Table 1 ijerph-20-00102-t001:** Characteristics of the participants.

	Total (N = 1027)	Frequency (%)
**Variables**		
**Age (years) mean (SD)**	25.8 ± 6.0	
**Age (years)**		
≤19	148	14.4
20–29	633	61.6
≥30	246	23.9
**Education level**		
None	650	63.3
Primary or more	377	36.7
**Wealth quintile**		
Lowest	205	20.0
Second	205	20.0
Middle	206	20.0
Fourth	205	20.0
Highest	206	20.0
**Geographical location**		
Semi-urban	590	57.4
Rural	437	42.6
**Occupation**		
Employed	353	34.4
Not employed	674	65.6
**Parity**		
Primipara	487	47.4
Multipara	540	52.6
**Marital status**		
Married	996	97.0
Not married	31	3.0
**Pre-existing illnesses or family history of hypertension**		
No	980	95.4
Yes	47	4.6

**Table 2 ijerph-20-00102-t002:** Alcohol consumption among pregnant women and associated factors in rural and semi-urban Burkina Faso 2019 (Crude, adjusted prevalence ratio, and 95% CI).

			Crude PR			Adjusted PR		
Variables	Total (N)	Prevalence (%)	PR	CI 95%	*p*	aPR	CI 95%	*p*
**All participants**	1027	10.2		[8.51–12.23]				
**Age**					0.28			0.13
≤19	148	10.1	1			1		
20–29	633	9.2	0.90	[0.51–1.59]		1.02	[0.58–1.77]	
≥30	246	13	1.28	[0.69–2.37]		1.61	[0.82–3.14]	
**Education**					0.04			<0.001
None	650	8.6	1			1		
Primary or more	377	13	1.51	[1.03–2.21] **		1.82	[1.27–2.62] **	
**Wealth quintile**					0.32			0.38
Low	205	11.7	1			1		
Second	205	10.2	0.87	[0.49–1.57]		0.87	[0.49–1.52]	
Middle	206	9.2	0.79	[0.43–1.44]		0.75	[0.42–1.32]	
Fourth	205	13.2	1.12	[0.65–1.95]		1.08	[0.65–1.82]	
Highest	206	6.8	0.58	[0.302–1.12]		0.62	[0.32–1.19]	
**Geographical location**					0.02			<0.001
Semi urban	590	8.1	1			1		
Rural	437	13.0	1.60	[1.09–2.35] **		1.94	[1.31–2.87] **	
**Occupation**					0.29			0.82
Employed	353	8.8	1			1		
Not employed	674	10.9	1.25	[0.82–1.90]		1.05	[0.68–1.63]	
**Parity**					0.87			0.83
Primipara	487	10.0	1			1		
Multipara	540	10.4	1.03	[0.70–1.51]		0.95	[0.62–1.47]	
**Pre-existing illnesses or family history of hypertension**					0.20			0.14
No	980	10.5	1			1		
Yes	47	4.3	0.40	[0.09–1.64]		0.36	[0.93–1.38]	

aPR = adjusted prevalence ratio, cPR = crude prevalence ratio, ** = significant association

**Table 3 ijerph-20-00102-t003:** Tobacco use among pregnant women and associated factors in rural and semi-urban Burkina Faso, 2019; (Crude, adjusted prevalence ratio, and 95% CI).

			Crude Prevalence Ratio			Adjusted Prevalence Ratio		
Variables	Total (N)	Prevalence (%)	cPR	95% CI	*p*	aPR	95% CI	*p*
**All participants**	1027	3.0		[2.1–4.3]				
**Age (years)**					<0.001			<0.001
≤19	148	2.0	1			1		
20–29	633	1.42	0.70	[0.19–2.59]		0.15	[0.04–0.50] **	
≥30	246	7.7	3.81	[1.13–12.87] **		0.62	[0.19–2.06]	
**Education**					0.005			0.20
Primary or more	650	0.8	1			1		
None	377	4.3	5.41	[1.64–17.80] **		2.04	[0.67–6.11]	
**Wealth quintile**					0.67			0.39
Low	205	1.9	1			1		
Second	205	2.4	1.25	[0.34–4.65]		1.04	[0.32–4.43]	
Middle	206	3.4	1.74	[0.51–5.95]		1.59	[0.53–4.84]	
Fourth	205	4.4	2.25	[0.69–7.31]		1.98	[0.64–6.14]	
Highest	206	2.9	1.49	[0.42–5.29]		2.51	[0.81–7.82]	
**Geographical location**					<0.001			<0.001
Semi-urban	590	0.3	1			1		
Rural	437	6.6	19.58	[4.67–82.04]		15.02	[3.12–72.39] **	
**Occupation**					0.02			0.30
Employed	353	1.1	1			1		
Not employed	674	4.0	3.53	[1.24–10.10] **		1.69	[0.63–4.55]	
**Parity**					<0.001			<0.001
Primipara	487	0.6	1			1		
Multipara	540	5.2	8.42	[2.56–27.69]		10.07	[3.48–29.14] **	
**Pre-existing illnesses or family history of hypertension**					0.19			0.82
No	980	2.9	1			1		
Yes	47	6.4	2.23	[0.68–7.35]		1.13	[0.38–3.37]	

aPR = adjusted prevalence ratio, cPR = crude prevalence ratio, ** = significant association

**Table 4 ijerph-20-00102-t004:** Inadequate physical activity among pregnant women and associated factors in rural and semi-urban Burkina Faso, 2019; (Crude, Adjusted prevalence ratio, and 95% CI).

			Crude Prevalence Ratio			Adjusted Prevalence Ratio		
Variables	Total (N)	Prevalence(%)	cPR	95% CI	*p*	aPR	95% CI	*p*
**All participants**	1027	79.4		[76.9–81.8]				
**Age**					0.25			0.03
≤19	148	74.3	1			1		
20–29	633	77.6	1.04	[0.84–1.28]		1.02	[0.92–1.12]	
≥30	246	87.4	1.17	[0.93–1.48]		1.1	[0.99–1.23]	
**Education**					0.05			0.14
None	650	75.2	1			1		
Primary or more	377	86.7	1.15	[1.00–1.33] **		1.04	[0.98–1.10]	
**Wealth quintile**					<0.001			<0.001
Low	205	73.7	1			1		
Second	205	62.9	0.85	[0.67–1.08]		0.90	[0.80–1.01]	
Middle	206	84.5	1.15	[0.92–1.43]		1.16	[1.06–1.27] **	
Fourth	205	88.3	1.19	[0.96–1.49]		1.25	[1.15–1.36] **	
Highest	206	87.9	1.19	[0.96–1.48]		1.16	[1.06–1.27] **	
**Geographical location**					<0.001			<0.001
Rural	590	59.9	1			1		
Semi-urban	437	93.9	1.57	[1.35–1.81] **		1.54	[1.42–1.66] **	
**Occupation**					0.08			0.17
Employed	353	86.1	1			1		
Not employed	674	75.9	0.88	[0.76–1.02]		1.04	[0.98–1.10]	
**Parity**					0.67			0.74
Primipara	487	80.7	1			1		
Multipara	540	78.3	0.97	[0.85–1.11]		1.01	[0.95–1.07]	
**Pre-existing illnesses or family history of hypertension**					0.16			0.14
No	980	80.3	1			1		
Yes	47	61.7	0.77	[0.53–1.11]		0.85	[0.68–1.05]	

aPR = adjusted prevalence ratio, cPR = crude prevalence ratio, ** = significant association

**Table 5 ijerph-20-00102-t005:** Insufficient consumption of fruit and vegetables among pregnant women and associated factors in rural and semi-urban Burkina Faso, 2019; (Crude, adjusted prevalence ratio, and 95% CI).

			Crude Prevalence Ratio			Adjusted Prevalence Ratio		
Variables	Total (N)	Prevalence (%)	cPR	95% CI	*p*	aPR	95% CI	*p*
**All participants**	1027	53.5						
**Age**					0.16			0.30
≤19	148	62.2	1			1		
20–29	633	53.9	0.87	[0.69–1.09]		0.92	[0.79–1.06]	
≥30	246	47.6	0.76	[0.58–1.01]		0.86	[0.72–1.04]	
**Education**					<0.001			0.01
Primary or more	650	37.7	1			1		
None	377	62.8	1.67	[1.38–2.02] **		1.16	[1.03–1.30] **	
**Wealth quintile**					0.98			<0.001
Low	205	52.2	1			1		
Second	205	55.1	1.06	[0.81–1.37]		0.90	[0.80–1.01]	
Middle	206	51.9	0.99	[0.76–1.30]		0.94	[0.83–1.06]	
Fourth	205	53.7	1.03	[0.79–1.34]		0.90	[0.79–1.03]	
Highest	206	54.8	1.05	[0.81–1.37]		1.19	[1.03–1.40] **	
**Geographical location**					<0.001			<0.001
Semi Urban	590	23.1	1			1		
Rural	437	94.7	4.11	[3.39–4.99] **		4.47	[3.85–5.18] **	
**Occupation**					0.11			<0.001
Employed	353	48.4	1			1		
Not employed	674	56.2	1.16	[0.96–1.39]		0.73	[0.66–0.80]	
**Parity**					0.01			0.40
Primipara	487	47.6	1			1		
Multipara	540	58.9	1.24	[1.04–1.46]		1.05	[0.93–1.19]	
**Pre-existing illnesses or family history of hypertension**					0.03			0.02
No	980	52.4	1			1		
Yes	47	76.6	1.46	[1.04–2.05]		1.17	[1.03–1.33] **	

aPR = adjusted prevalence ratio; cPR= crude prevalence ratio, ** = significant association

**Table 6 ijerph-20-00102-t006:** Factors associated with the presence of more than one cardiovascular lifestyle risk factor in rural and semi-urban Burkina Faso, 2019.

Variables	Total (N)	Prevalence (%)	cPR (IC 95%)	*p*	aPR (IC 95%)	*p*
**All participants**	1027	50.9				
**Age**				0.001		0.001
≤19	148	41.2	1		1	
20–29	633	48.8	1.18 [0.96–1.46]		1.06 [0.89–1.25]	
≥30	246	62.2	1.51 [1.22–1.87]		1.26 [1.03–1.53] **	
**Education**				0.001		0.08
None	650	42.8	1		1	
Primary and more	377	65	1.52 [1.35–1.71] **		1.09 [0.99–1.20]	
**Wealth quintile**				0.72		
Low	205	51.7	1			0.01
Second	205	47.8	0.92 [0.76–1.12]		1.10 [0.96–1.26]	
Middle	206	51.9	1 [0.83–1.21]		1.07 [0.93–1.23]	
Fourth	205	54.2	1.05 [0.87–1.26]		1.23 [1.06–1.42] **	
Highest	206	49.0	0.95 [0.78–1.15]		0.94 [0.79–1.11]	
**Geographical location**				0.001		0.001
Rural	590	14.9	1		1	
Semi-urban	437	77.6	5.22 [4.15–6.56] **		5.46 [4.34–6.88] **	
**Occupation**				0.13		0.001
Employee	353	54.1	1		1	
None	674	49.3	0.91 [0.81–1.03]		1.31 [1.18–1.44] **	
**Parity**				0.03		0.68
Primipara	487	54.4	1		1	
Multipara	540	47.8	0.88 [0.78–0.99]		0.98 [0.87–1.09]	
**Pre-existing illnesses or family history of hypertension**						0.16
Yes	980	29.8	1		1	
No	47	58.9	1.98 [1.27–3.08] **		1.30 [0.90–1.87]	

cPR = Crude Prevalence Ratio; aPR = adjusted Prevalence Ratio, ** = significant association

## Data Availability

Not applicable.

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
