# Peer review of "Prevalence and Factors Associated with Cardiovascular Lifestyle Risk Factors among Pregnant Women in Burkina Faso: Evidence from a Cross-Sectional Study"

_ijerph, 2022, doi:10.3390/ijerph20010102_

Round 1

Reviewer 1 Report

This study on risk factors of CVD in a pregnant women is based on a large representative sample from rural and urban areas of Burkina Faso. I have no concerns with the study design which seems to be adequate.

However, I have several questions concerning the statistical analysis and the way that results are presented. There seem to be several errors in the tables which have to be corrected before a final review is possible.

In detail:

- What was the reason that you chose Poisson regression with GEE instead of the usual logistic regression? References 10 and 28 are not helpful to most readers.  Which STATA procedure did you use for the calculations? The sample size formula is printed incorrectly and looks like the formula for logistic regression and odds ratios.

- There is a confusion with the tables: line185 contains a wrong reference to Table 1, then there are two Tables 2. Why are the variables for fruit and vegetable consumption combined in the second, but not the first table? If there is a high correlation, they should be combined in all analyses. In addition, it is not clear whether these are counted as one or two RF in the analysis for "at least 2 RF". The order of predictor variables should be made identical in both tables.

- The first Table 2 contains numeric errors: for example, the absolute numbers in the line "not married" are much too large. There are more implausibilities, so the whole table has to be checked.

- line 160: what is meant by "we reported a global p-value"?

- There are several typos (e.g. fever instead of fewer) and punctuation errors.

- In the abstract as well as in the text and tables, percentages should be given with one decimal instead of two) for better readability. For ORs or PRs two decimals are ok.

Author Response

We would like to thank you for your valuable comments. We have addressed all the comments and produce point by point responses in the present document. The text with red color indicates or explains the changes we operated in the revised manuscript. The text with green font is the new text inserted in the revised manuscript. In the manuscript modifications are highlighted by tracked change.

Best regards

Reviewer 2 Report

Dear authors,

The paper "High prevalence of cardiovascular lifestyle risk factors among pregnant women in Burkina Faso:  evidence from a cross sectional study" is interesting. My comments are as follows:

1. Did you account for differences in maternal race? Or were all participants from African origin?

2. Did you consider any pre-existing illnesses or family history of cardiovascular diseases in this study?

3. Data on maternal smoking, and alcohol consumption were assessed in a self-administered questionnaire. It is a not verifiable survey and could introduce bias which should be discussed.

Author Response

We would like to thank you for your valuable comments. We have addressed all the comments and produce point by point responses in the present document. The text with red color indicates or explains the changes we operated in the revised manuscript. The text with green font is the new text inserted in the revised manuscript. In the manuscript modifications are highlighted by tracked change.

Best regards.

Round 2

Reviewer 1 Report

thank you for carefully revising the manuscript. the tables have clearly improved.